# Barriers and Facilitators to Point-of-Care Ultrasound Use in Rural Australia

**DOI:** 10.3390/ijerph20105821

**Published:** 2023-05-14

**Authors:** Annie C. Arnold, Richard Fleet, David Lim

**Affiliations:** 1College of Medicine and Public Health, Flinders University, Bedford Park, SA 5042, Australia; arno0068@flinders.edu.au; 2Family and Emergency Medicine, Universite Laval, Quebec City, QC G1V0A6, Canada; richard.fleet@fmed.ulaval.ca; 3Translational Health Research Institute, School of Health Sciences, Western Sydney University, Campbelltown, NSW 2560, Australia

**Keywords:** diagnostic imaging, paediatric diagnostic imaging, patient transfers, point-of-care ultrasound, rural medicine, telemedicine

## Abstract

This study explores the barriers and facilitators to point-of-care ultrasound (POCUS) use and adoption in rural healthcare since POCUS is a useful resource for rural clinicians to overcome the challenges associated with limited on-site clinical support, such as limited diagnostic imaging services and infrastructure. A qualitative descriptive study was employed, interviews with ten rural clinicians were conducted, and the data were analysed using the Walt and Gilson health policy framework to guide interpretation. Barriers include a lack of standardised training requirements, the cost of the devices and challenges recouping the costs of purchase and training, difficulty with the maintenance of skills, and a lack of an effective method to achieve quality assurance. Coupling POCUS with telemedicine could address the issues of the maintenance of skills and quality assurance to facilitate increased POCUS use, leading to positive patient safety and social and economic implications.

## 1. Introduction

In many countries, rural and remote populations continue to experience significant healthcare disadvantages due to challenges relating to geographic spread, low population density, and infrastructure limitations. In Australia, for instance, both the Australian Institute of Health and Welfare (AIHW) [1] and the recent New South Wales report on rural health outcomes [2] highlighted concerns about the persistent disparity between the health of regional residents compared to their city-based counterparts. After adjusting for age, the AIHW found an increased burden of disease, chronic conditions, and premature and potentially avoidable deaths with increasing remoteness, suggesting that health outcomes are inextricably linked to a patient’s rurality [1]. Meanwhile, the New South Wales report presents an inverse picture of healthcare whereby some of the most vulnerable populations are the most disadvantaged by healthcare accessibility, facing significant travel requirements, increased costs, and delays in accessing care [2].

Contributing to the challenge of rural healthcare delivery includes limited access to diagnostic imaging equipment and services [3,4]. Point-of-care ultrasound (POCUS) is a widely available technology that provides clinicians with immediate bedside diagnostic imaging. The portability and versatility of POCUS make it particularly well-suited to rural medicine as a resourceful way to overcome challenges associated with limited on-site clinical support. POCUS has the potential to improve patient outcomes by allowing clinicians to tailor management to a known diagnosis thus enabling patients to receive definitive treatment earlier than they would otherwise [5,6]. Additionally, POCUS can simplify rural patients’ journey through the healthcare system, with initial assessment, diagnosis, and management able to be provided in a single consultation [5,6]. The applications of POCUS for rural medicine are numerous; however, challenges associated with implementation and maintenance need to be addressed. This research aims to explore the current understanding of the barriers and enablers to POCUS use in rural healthcare practice.

## 2. Methods

A qualitative descriptive study was employed. Semi-structured interviews were conducted between January 2021 and December 2021 with a purposive sample of rural clinicians who had been working in rural or regional South Australia for at least 12 months. The following agencies assisted with the recruitment process: Flinders University Rural Clinical School, Rural Doctors Association Australia, and the Rural Doctors Workforce Agency. Interviews were conducted in person or via video-conferencing platforms, were audio-recorded, and lasted between 30–60 min. The interviews were transcribed verbatim and returned to the participants for member-checking prior to analysis. Data saturation was considered reached when no new information was forthcoming within subsequent interviews [7]. Thematic analysis of the transcripts was performed; the initial analysis was centred on the inductive identification of themes whereby emerging conceptual categories were identified and coded. Overlapping codes were consolidated and grouped into categories. Walt and Gilson’s health policy analysis framework [8], which appreciates the complex, iterative process of health policy development and the need to address both upstream and downstream policy content, was utilised to guide the data interpretation and explore the interconnected elements important in health policy. An audit trail of the coding process, a reflective journal kept by A.C.A., member checking of de-identified transcripts, multiple coding (A.C.A. and D.L., further discussion with R.F.), and peer review (through the use of an informed insider) were used to maintain research rigour.

## 3. Results

Ten interviews [C01–C10] with practising rural clinicians were conducted, with representation across gender, years in practice, rurality, and specialty as described in Table 1. Based on data from the National Health Workforce Dataset 2020 [9], the sample reflected the current gender and age demographics of medical practitioners working in rural and remote areas. The Modified Monash Model 2019 rurality classification was used; the sample was roughly representative of the current distribution of the South Australian rural workforce [9].

Four core themes were derived and are summarised in Table 2 below.

### 3.1. POCUS as a Rural Health Initiative

The rurality of practice was unanimously identified as a significant contextual factor influencing the use of POCUS [C01–C10]. POCUS was highly valued in locations with limited on-site clinical support [C03, C04, C06, C08, and C09], with POCUS “providing a dimension of understanding that you wouldn’t have otherwise” [C08]. The participants described the value of POCUS use in after-hours practice when they were unable to access formal diagnostic imaging services locally [C01 and C03–C08]. Although most of the participants reported easy access to formal radiology during business hours [C03–C05 and C08], clinicians working in increasing rurality with increasingly limited access to healthcare infrastructure described POCUS as “invaluable” [C01]. In such scenarios, POCUS use influenced patient outcomes by helping to “know the acuity of the patient” [C06] and differentiate between causes of a shock to allow earlier definitive management, with notable examples being using POCUS to diagnose conditions such as pericardial tamponade [C08], splenic laceration [C05], and pneumothorax [C04]. The additional information POCUS provides enabled rural clinicians to make a “more informed referral” [C08] to a tertiary center and management prior to transfer to be more appropriately tailored to the patient [C04, C06, and C08]. Ruling out emergent conditions at the bedside to prevent the need to transfer a patient for the purpose of diagnostic imaging was an application of particular benefit in a rural environment [C04–C06 and C08], with a common example being assessing bleeding in early pregnancy to exclude a life-threatening ectopic pregnancy [C04, C05, and C09].

### 3.2. The Influence of Actors in Driving POCUS Use

A lack of standardised training requirements leaves the decision to pursue POCUS to the individual motivations of the clinician and the benefits POCUS provides to their practice. Individual motivation was cited to be based on a desire to upskill and be able to practise more independently [C01, C04, and C06] or motivated by a patient-centered approach to care [C01, C04, C06, and C09]. The ability to diagnose and manage patients in a single consultation was an attractive application [C01, C02, and C06]. The examples provided were imaging and aspirating breast lumps [C02 and C06] and performing intra-articular injections [C01 and C06], where ordinarily patients would require formal diagnostic imaging provided outside the primary care setting. Preventing patients from needing multiple consultations within the healthcare system increased patient satisfaction and improved the doctor–patient relationship [C05, C06, and C09].

### 3.3. Issues with Implementing and Maintaining POCUS in Rural Practice

Participants identified various factors relating to training as significant, including access [C03, C06, C08, and C10], cost [C01, C08, and C10], and the standardisation of training requirements [C03 and C06]. Training courses for bedside ultrasound are widely available through a number of independent providers, and many participants were able to access training through professional development grants and other funding avenues [C03, C04, C06, C07, and C09]. Access was limited for the participants due to requirements to travel, often interstate, to attend such courses [C06, C08, and C10]. As such, a lack of time and the cost of travel were identified as significant barriers [C04, C06, and C10]. It was recognised that the Rural Doctors Workforce Agency (RDWA), in conjunction with local education provider LearnEM, is already addressing accessibility issues for South Australian practitioners by funding and providing training workshops that have been accredited by the Australasian Society for Ultrasound in Medicine [C04, C05, and C10]. The RDWA has identified funding as the limiting factor in scaling the provision of these workshops to meet demand and that further investment in the provision of these local workshops should be considered [C10].

The issues relating to the maintenance of skills were a significant barrier, with the participants reporting a rapid loss of skills and challenges in integrating learning from training into everyday practice [C03, C04, and C10]. As one participant explained, “One of the barriers is perfecting the techniques in your own practice. There’s a real drop off from finishing the course to implementing the skills in practice” [C10]. The ability to practise regularly soon after training was a logistical challenge requiring time on the clinician’s end, plus suitable patient presentations and clinical scenarios where POCUS could be applied [C03, C04, and C06–C10].

The lack of an effective, structured method to achieve quality assurance was identified as an issue [C04–C06]. Clinicians’ confidence in their skills of image production and interpretation had a significant influence on their decision to use POCUS, including assessing whether a patient required transfer to a tertiary center—previously identified as one of the most valuable applications of POCUS [C04–C06 and C08]. As one participant explained, “The tricky thing is those decisions are so important to get right, and if I’m only doing it intermittently, I don’t feel confident enough in my ability to be scanning and excluding that emergency entirely” [C04]. The ability to corroborate the interpretation of an image with another experienced user or access remote support from an expert radiographer were identified as ways to address this [C03, C04, C06, and C10].

### 3.4. Suggestions for a New POCUS Model for Rural Healthcare

It was proposed that establishing a more standardised approach to ultrasound certification for rural trainees could increase the use of POCUS [C03, C06, and C10]. The participants described a lack of consistency in training experiences and workplace requirements as having an impact on users’ confidence [C06 and C07], as they were unable to standardise their level of expertise and had “little understanding of what it takes to be proficient” [C07]. Key suggestions were to include POCUS as a core competency for all rural trainees and include basic POCUS education in medical schools and education programs for junior doctors prior to specialisation [C01, C02, C03, C05, and C06].

Establishing a financial incentive for POCUS in general practice was another prominent subtheme. Although the participants recognised the benefit of having a personal POCUS device for their practice, the cost was the limiting factor [C03, C06, C09, and C10]. The price point has decreased significantly over the past few years, with the current cost of an entry-level ultrasound being approximately AUD 4000 [10], but the participants reported difficulty recouping these costs given that currently, they are unable to bill to reflect that an appointment involved a point-of-care scan. When coupled with the fact that the use of ultrasound can increase the consultation time [C05 and C09], general practitioners do not receive any compensation for their time or efforts. Introducing new Medicare item numbers to cover ultrasound was suggested as a solution to help incentivise clinicians to pursue ultrasound and help financially compensate for the cost and registration of the device and the added costs and time required for training [C01 and C09].

## 4. Discussion

The key themes identified in this study suggest that increasing POCUS use has patient safety, social, and economic implications. From a patient safety perspective, the scope for POCUS in rural practice is significant. Both the participants’ responses and the published literature support that POCUS is a useful adjunct that can improve diagnostic accuracy and provide a definite diagnosis earlier [5,6,11,12,13]. It is reasonable to assume that the diagnostic benefits of POCUS may translate into improved patient outcomes, however, research in this area is lacking. The positive impact of POCUS use on patient satisfaction and the doctor–patient relationship should also be considered as having an influence on patient outcomes [5,6,14]. The potential for POCUS use to reduce healthcare costs by helping to clarify patient transfer decisions is significant [15]. Although no studies have been conducted in Australia, a recent New Zealand study demonstrated that POCUS use resulted in de-escalation of the level of care, reducing hospital admissions and inter-hospital transfers [13], with transfers being one of the most resource-intensive decisions clinicians must make [16]. Even a modest reduction in patient transfers has the potential to result in significant savings for both individual patients and the healthcare system. As another incentive, research on attracting and retaining a rural workforce identified that clinical support was a significant factor for junior doctors [17,18]. Ensuring that rural locations are well equipped with POCUS as a diagnostic tool may help provide additional support for junior doctors, thus helping to address rural workforce issues.

The following recommendations are made in line with the policy framework. Suggested upstream policy changes are to standardise POCUS exposure and training for the future workforce. The present study identified value in the introduction of basic POCUS skills into medical school curriculums and ensuring that POCUS training is provided to all rural trainees. The Australian College of Rural and Remote Medicine already recognises the value of having POCUS in the armamentarium of rural generalists, with POCUS thus included as a core competency in the advanced rural generalist curriculum [19]. Another key recommendation is to provide a financial incentive through the introduction of Medicare item numbers covering the use of POCUS to compensate for the costs associated with training and the purchase of a device.

Rapid loss of skills following training is a well-known phenomenon [20,21]. Coupled with the lack of standardised training requirements, this can lead to significant inter-operator variability in POCUS interpretation, and thus downstream policy recommendations address the maintenance of skills and quality assurance. Telemedicine has seen a dramatic increase in utility over the last three years during the COVID-19 pandemic, and there is a resultant readiness of infrastructure required to support telemedicine [22]. Advances in the visual acuity of both POCUS devices and videoconferencing platforms allow for images produced with POCUS to be transmitted in real-time to a tertiary center where an expert can assist with image production and interpretation. The idea of coupling POCUS with telemedicine to transmit images in real-time for remote diagnostic support is not novel; teleultrasound is already being used to support clinicians in many resource-limited settings [15,23,24]. The South Australian Digital Telehealth Network already provides after-hours clinical support to country health services, and the SAAS MedSTAR specialist emergency medical retrieval service for the state of South Australia has a videoconferencing service where doctors working in rural emergency departments can receive specialist consultant advice about patient management [25]. This service has already been used in a makeshift capacity for teleultrasound, demonstrating that although not currently routine or standardised, the current infrastructure supports this application.

One of the limitations of this study is that whilst the small number of participants are all clinicians practising in rural and regional South Australia, this may have limited the transferability of the study findings to other jurisdictions and remote settings; the sample did, however, capture decades of experience from a range of specialties across all rural indices. By virtue of the participant’s vantage as clinicians with personal experience in the use of POCUS, the study was able to achieve an in-depth and nuanced understanding of clinician experiences regarding the challenges associated with implementing POCUS, therefore there is credibility regarding the robustness of the information presented in this study. Selection bias may be present, given the purposive sampling methodology. Further research that builds on the findings of this study may be warranted to fill the gaps identified.

## 5. Conclusions

The findings contribute to the current understanding of the role of POCUS and the most salient barriers to its widespread use in rural South Australia, identifying practical learning which could be applied to achieve widespread and sustained use. The maintenance of skills, quality assurance, and standardised exposure require economic, policy, and system changes to achieve the patient-centered benefits of POCUS.

## Figures and Tables

**Table 1 ijerph-20-05821-t001:** Characteristics of the study participants.

Demographic	Number	%
Male	6	60
Female	4	40
Years in practice		
5–10 years	4	40%
10–20 years	1	10%
20–30 years	2	20%
>30 years	3	30%
Rurality (Modified Monash Model Category)		
MM 3 (large rural towns)	3	30%
MM 4 (medium rural towns)	5	50%
MM 5 (small rural towns)	3	30%
MM 6 (remote communities)	1	10%
MM 7 (very remote communities)	2	20%

**Table 2 ijerph-20-05821-t002:** Summary of relevant direct quotations from participants represented within the Walt and Gilson framework [8].

**Context**	
Location and rurality	“The only thing they have there is X-ray services available half a day a week, and they have no other radiology available, and so for a place as remote as that, bedside ultrasound would be invaluable because you have nothing else, so I mean it’s great that you can at least get an ultrasound there next to you.” [C01] “We have to do our own ultrasound after hours in lieu of X-rays and CTs and formal ultrasound. So that’s where it’s very useful. You get the benefit of ultrasound when there’s no other imaging after hours.” [C08]
Social context	“Being confident in how to manage people prior to transfer, and potentially better patient outcomes in that respect. They’d be in better condition by the time they get there.” [C04] “It helps with knowing the acuity of the patient. So, it helps with management as well, and communicating with people back in the city.” [C06] “You could make a more informed referral. You can talk in a more informed way to the person on the other end. So, it makes the referral process more specific, and when it needs to be urgent, you can often get the people on the other end to respond more appropriately if they know what you are dealing with.” [C08]
Political context	“That was purely out of interest because I wanted to learn how to do things a bit more independently and do procedural things independently, as opposed to having to find someone to do something for me or refer things on.” [C01] “I could see its use in the emergency setting. I think I did it because I just wanted to increase my confidence in emergency settings.” [C04] “It’s one of those things, you get no incentive for it. Other than just wanting to be better at just doing things.” [C06]
**Content**	
Standardised exposure and training	“If it was something that was sort of better accessible or something that we were potentially taught in medical school even that will be quite useful.” [C01] “It is becoming a requirement, so that trainees coming through the emergency medicine college have to become competent in bedside ultrasound.” [C08]
Provision of equipment	“It’s reliant on your local health service getting on board and recognising that this should be standard.” [C05] “You put the onus back on the health service. You know, is this standard for a rural facility of this size? Really, we should have 2 ultrasound scanners, you know in case one breaks down.” [C06]
Financial incentive	“I think if there was an MBS (Medicare Benefits Scheme) item associated with bedside ultrasound, you’d find the use would skyrocket, if there was an MBS item”. [C01]
Suggested policy changes	“And for resus as well, I want to get everyone to use the scanner for resus. So, I’m putting it in, you know you go through your ABC [Airway, Breathing, Circulation] approach, and then you add in ultrasound. By just putting things into policy, you can change things as well.” [C06]
**Actors**	
Clinician driven	“That was purely out of interest because I wanted to learn how to do things a bit more independently and do procedural things independently, as opposed to having to find someone to do something for me or refer things on”. [C01] “I could see its use in the emergency setting. I think I did it because I just wanted to increase my confidence in emergency settings.” [C04] “It’s one of those things, you get no incentive for it. Other than just wanting to be better at just doing things.” [C06]
Patient driven	“Particularly as our patient population is getting bigger and bigger it’s actually getting harder to find veins.” [C04]
Workplace driven	“That course was part of an emergency course that we do every two or three years to enable us to be more confident managing emergencies.” [C04]
**Process**	
Training	“I definitely think training is a big part of it, and training people earlier in their medical careers”. [C01] “I think the cost of training is pretty much covered by a lot of the grants that we do get, and I think the government are very good at providing that.” [C04] “There’s a large range in skills, mainly because the training for it is off your own back, it’s all ad hoc. Like there’s no formalisation of it. There’s no way to standardise your level of expertise.” [C06]
Access to equipment	“We actually have just the one machine in the hospital. There’s this weird concept that because it’s a rural facility you only need one, it doesn’t matter what size.” [C06] “We don’t have a dedicated ultrasound machine in the ED [Emergency Department]. And that becomes quite annoying because you have to fight the labour ward for the ultrasound machine, and theatre too. And it’s not available for those places all the time.” [C07]
Cost	“I think the other thing is definitely the cost to the doctor, the cost of owning the machine. And obviously, you can’t bill through MBS (Medicare Benefits Scheme) for bedside ultrasounds, so there’s no real financial kickback for it”. [C01] “The main difficulty I think, is the cost, the cost of the machines. I think that is where people struggle because there is no money in it. You know, you have no way of billing if you’ve added in an ultrasound, so unfortunately you don’t have a way of recuperating that cost, because all you can bill it as is a standard consult.” [C09]
Maintenance of skills	“One of the barriers that people run into that have done training, whether it be the 5 day or the 2 day or whatever training they have done, they then need to perfect the techniques in their own practice, and make sure that they are doing that enough that keep the skills that they’ve learned up, and they often say there’s a real drop off from when they finish the course to when they can implement their skills in practice.” [C10]
Quality assurance	“I think that’s one of the reasons people don’t pick it up, is they’re too scared because they do worry about their skills. So, then you end up getting this fear of interpretation.” [C06] “Quality assurance, so what things are in place for you to consolidate your skills or, you know, just make sure you’re doing it properly. Because for us, like I’ve said already, this is not something that we’re doing very often.” [C05]

## Data Availability

The data presented in this study are available from the corresponding author upon reasonable request. The raw data are not publicly available due to the conditions of institutional ethics approval.

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
