# Peer review of "Barriers and Facilitators to Point-of-Care Ultrasound Use in Rural Australia"

_ijerph, 2023, doi:10.3390/ijerph20105821_

Round 1

Reviewer 1 Report

Thank for the opportunity to review this important paper, that captures useful insights from a clinician perspective on the utility of point of care ultrasound in rural practice. I offer the below comments to assist the authors with further developing their manuscript in preparation for publication.

p. 2 L 49 I don’t think the ‘the’[ before current is needed

p. 2 L 53-4 rather than the research topic, I think re-state so does not read like a template.

p. 2 L 64 – I think in terms of presenting information in the order in which actions in the process occurred may mean that this line about inductive coding should come before the previous line about applying the framework (deductive) and perhaps useful to include further explanation re used inductive first and then applied framework to identify themes that may not have been articulated in the framework.

p. 2 L66-67 could you provide more detail about how an audit trial, reflective journalling, member checking etc. were used for research rigor?

p. 3 L 78 the idea that themes ‘emerge’ in the process of thematic analysis has been critiqued for some time and the authors may like to refer to more recent methodological literature in this space.

p. 7 L 190-191 – this line about the framework may be more useful earlier on in the methodology section.

p. 8 L 220 I wonder about the appropriateness of using generalisability as a marker of quality/point of ‘limitation’ for the study as this is not the aim of qualitative research. Linda Finlay has written some useful papers on this matter e.g. https://doi.org/10.1177/030802260606900704 which may help the authors re-consider this. Relatedly, rather than larger sample size per sae it could be more a mixed method study that would build on the findings of this research and fill the gaps identified?

Author Response

Thank you for the constructive feedback for improvement, we really appreciate it.

Response to Reviewer 1's comments:

  • p2 L49: I don't think "the" before "current ..." is needed.

Response: we have now removed the word as recommended.

  • p2 L53-4: restate so it does not read like a template

Response: we have now amended the sentence using track changes to avoid duplication.

  • p2 L64: I think in terms of presenting information in the order in which actions in the process occurred may mean that this line about inductive coding should come before the previous line about applying the framework

Response: thank you for the insightful comment. We have now amended the sentence accordingly.

  • p2 L66-7: could you provide more details about how an audit trail, reflective journalling, member checking etc were used for research rigour?

Response: thank you for the opportunity to provide further information. We have now elaborate on the sentence as suggested.

  • p3 L78: the idea that themes "emerge" in the process of thematic analysis has been critiqued for some time.

Response: we have rephrased the sentence that the themes are derived from the data.

  •  p7 L190-1: this line about framework may be more useful earlier on in the methodology section

Response: thank you for the comment. We have now moved the sentence up to the methods section.

  • p8 L220: I wonder about the appropriateness of using generalisability as a marker of quality.

Response: thank you for the comment, we have now replaced the term with transferability and noted that further studies are warranted.

Reviewer 2 Report

The topic is timely and important. The sample size, even for a qualitative study, is small (n=10) and while I understand the relevance to the extremely rural contexts described I think that the use of POCUS is relevant in most contexts outside of urban centers so it should have been possible to recruit more clinicians than you did.

I would have liked to have seen the interview guide....it seems, by the length of the interviews, that the authors failed to dig deep into the issues surrounding the use of POCUS and possible novel solutions.

Some of the conclusions, though logical, do not seem to flow from the data (ie curriculum in medical school). I can think of other potential solutions that might be more feasible.

Author Response

Thank you for the constructive comment and affirmation of the importance of this work.

Comments: The topic is timely and important. The sample size, even for a qualitative study, is small (n=10) and while I understand the relevance to the extremely rural contexts described I think that the use of POCUS is relevant in most contexts outside of urban centers so it should have been possible to recruit more clinicians than you did.

Response: This was a coursework medical student’s research project thus we were restricted by the timeframe when we are needed to complete the research in order for the student (AA) to graduate. (We have acknowledged the contribution of this research to AA’s degree in the Conflicts of Interest section of the manuscript). The recruitment was assisted by the University’s Rural Clinical School, and two other peak bodies in rural and remote Australia i.e. Rural Doctors Association Australia and the Rural Doctors Workforce Agency. We did anticipate a greater number of participants. The actual low number may be the fact that when the recruitment and interview were scheduled in January – December 2021, South Australia was confronted by its responses to the COVID-19 pandemic. The hard border closure and strict quarantine measures in South Australia were reinstated in June 2021, the suppression phase was only lifted in late November 2021. We have acknowledged the limitations of this study, including the small number of participants (p7 L219 revised, track changes). We noted in the demographic data of the participants, the gender, age and rurality of the participants were comparable to the national health workforce data (p2 L75-9).

Comments: I would have liked to have seen the interview guide....it seems, by the length of the interviews, that the authors failed to dig deep into the issues surrounding the use of POCUS and possible novel solutions.

Response: The interview guide is now attached.

Comments: Some of the conclusions, though logical, do not seem to flow from the data (ie curriculum in medical school). I can think of other potential solutions that might be more feasible.

Response: Thank you for the comment. We previously published a commentary on POCUS training in rural generalism training and we also made reference to the Australian College of Rural and Remote Medicine’s position on POCUS in the Discussion. Thus, the recommendation of the medical school curriculum was proposed so that upstream exposure to POCUS (in the medical school curriculum) could be better aligned with the downstream fellowship training. If the Reviewer is willing, we would like to follow-up on opportunities for future research on this topic after the manuscript is published, so that we can benefit from the Reviewer’s further wisdom and experience. We thank the Reviewer for his/her/their kind consideration.
